# Impact of hepatopathy in pediatric patients after surgery for complex congenital heart disease

Torben Kehl[1]☯*, Daniel Biermann[2]☯, Andrea Briem-Richter[3], Gerhard Schoen[4], Jakob Olfe[1], Joerg S. Sachweh[2], Lutz Fischer[5], Hansjoerg Schaefer[6], Rainer Kozlik-Feldmann[1], Urda Gottschalk[1]

1 Department for Pediatric Cardiology, University Heart & Vascular Center Hamburg, University Medical Center Hamburg-Eppendorf, Hamburg, Germany, 2 Cardiac Surgery for Congenital Heart Disease, University Heart & Vascular Center Hamburg, University Medical Center Hamburg-Eppendorf, Hamburg, Germany, 3 Department of Pediatrics, University Medical Center Hamburg-Eppendorf, Hamburg, Germany, 4 Institute of Medical Biometry and Epidemiology, University Medical Center Hamburg-Eppendorf, Hamburg, Germany, 5 Department of Visceral Transplantation, University Medical Center Hamburg-Eppendorf, Hamburg, Germany, 6 Institute of Pathology, University Medical Center Hamburg-Eppendorf, Hamburg, Germany

☯ These authors contributed equally to this work.
* t.kehl@uke.de

**Data Availability Statement:** All relevant data are within the paper and its Supporting Information files. The analysis was made with R and SPSS. An uploaded supporting zip file include a xlsx

## Abstract

Patients undergoing complex pediatric cardiac surgery in early infancy are at risk of postoperative secondary end-organ dysfunction. The aim of this study was to determine specific risk factors promoting the development of peri- and postoperative hepatopathy after surgery for congenital heart disease. In this retrospective study, we identified 20 consecutive patients operated between 2011 and 2019 from our institutional cohort who developed significant postsurgical hepatic dysfunction. These patients were compared to a control group of 30 patients with comparable initial cardiac conditions and STS-EACTS risk score. Patients who developed hepatopathy in the intensive care unit have chronic cholestasis and decreased liver synthesis. The impact of postoperative hepatopathy on morbidity was marked. In six patients (30%), liver transplantation was executed as ultima ratio, and two (10%) were listed for liver transplantation. The overall mortality related to postoperative hepatopathy is high: We found nine patients (45%) having severe hepatopathy and mostly multiple organ dysfunction who died in the postoperative course. According to risk analysis, postoperative right and left heart dysfunction in combination with a postoperative anatomical residuum needing a re-operation or re-intervention in the postoperative period is associated with a high risk for the development of cardiac hepatopathy. Furthermore, postoperative complications (pleural effusion, heart rhythm disorders, etc.), postoperative infections, and the need for parenteral nutrition also raise the risk for cardiac hepatopathy. Further investigations are needed to reduce hepatic complications and improve the general prognosis of such complex patients.

datasheet (corrected final version), analysis results in R as pdf file and SPSS analysis with corresponding files (corrected .sav and .sps file). A part of results of the statistical analysis are provided in German (pdf file); translation can be provided if required. The R source code on which the calculation is based can also be provided if required.

**Funding:** None of the authors received specific funding for this work.

**Competing interests:** The authors have declared that no competing interests exist.

# 1. Introduction

Congenital heart disease (CHD) is the most common congenital defect occurring in approximately 1% of all live births (excluding bicuspid aortic valve disease) [1]. The overall survival of CHD patients improved dramatically over the last decades, with grown-up congenital heart (GUCH) patients being alive now exceeding the number of children with CHD [2, 3]. The immense improvements in survival as a result of new surgical techniques, hybrid and catheter approaches, and advancements in intensive care management have put the focus of attention and research on possible complications related to CHD patient care. Patients with complex CHD defects and especially those, who need extensive heart surgery in early infancy, are still at high risk for increased postoperative morbidity and mortality.

Postoperative low cardiac output, affecting up to 25% of neonates and young infants after cardiac surgery and associated systemic inflammatory response syndrome (SIRS) following cardiopulmonary bypass (CPB) surgery, can trigger the risk for secondary organ dysfunction [4]. Neurological impairment and mental retardation in CHD patients are widely described [3, 5–7]. Renal dysfunction requiring temporary dialysis (either peritoneal dialysis or hemofiltration) to support and improve postoperative renal function is well outlined, too [8–10]. Not being in focus for a long time, the liver is at risk for perioperative acute and chronic complications, maybe more than expected. Liver dysfunction in failing Fontan circulation is uniformly known [11–13]. The underlying mechanisms of hepatic dysfunction are likely multifactorial. It is possibly a result of central venous congestion and low cardiac output resulting in tissue hypoxemia [14]. These mechanisms apply in chronic as well as in acute heart failure [13]. In addition, secondary sclerosing cholangitis (SSC) first observed in critically ill patients suffering burns or polytrauma newly also reported after cardiothoracic surgery associated with prolonged intensive care unit stay is characterized by biliary obstruction, cholestasis, and bile duct necrosis rapidly progressing to cirrhosis. It is becoming increasingly more diagnosed and described in recent literature [15–17]. A main pathomechanism of SSC might be a prolonged postoperative low cardiac output leading to insufficient oxygen delivery to the biliary epithelium, which is much more susceptible to ischemia than hepatocytes having dual blood supply [18, 19].

In this retrospective analysis, we focus on patients who developed severe liver dysfunction early in the postoperative course after complex cardiac surgery. The patients revealed cholestasis as well as elevated liver enzymes (alanine transaminase and aspartate transaminase) and impaired liver synthesis (limited coagulation factors and hypalbuminemia). This so-called "cardiac hepatopathy" complicates the postoperative course and is associated with morbidity and mortality. This study sought to evaluate specific perioperative risk factors, which can promote liver dysfunction postoperatively.

# 2. Methods and patients

## 2.1 Patients

All authors obeyed the Declaration of Helsinki. The local ethics committee waived approval of the study due to the retrospective and anonymous analysis performed (WF-055/20; ethics committee of the medical association Hamburg, Germany). Patient data was de-identified before our group had access to the data. Consequently, no informed patient consent was needed.

We identified and included 20 consecutive patients within our institutional cohort who met the criteria for cardiac hepatopathy (see 3.2) following pediatric cardiac surgery between 2011 and 2019 (mean center volume: 280 pediatric cardiac operations per year). These patients were retrospectively analyzed and compared to a control group consisting of 30 patients with a

comparable initial cardiac anatomy and pathophysiology but who did not develop postoperative hepatic dysfunction. To the best knowledge, our sample can be considered representative of a population with complex congenital disease.

The follow-up period was four months up to four years.

Patients were examined for underlying genetic abnormalities and syndromes in case of clinical suspicion.

## 2.2 Definitions

Cardiac hepatopathy was defined based on published definitions as persistent postoperative cholestasis with an elevation of specific laboratory cholestasis parameters (bilirubin, gamma-glutamyltransferase (gGT)), impaired liver synthesis (reduced production of albumin and/or coagulation factors) with or without liver sonographic and histopathological changes in the postoperative course [13, 20, 21].

Congenital heart defects were classified, according to Moss & Adams' Heart Disease [22]. Estimating the surgical risk related to the specific heart defects and their planned surgical procedure, the STS-EACTS mortality categories (1–5) were applied [23]. Right heart dysfunction was defined by repetitive measurements for the right ventricle function using tricuspid annular plane systolic excursion (TAPSE) on serial postoperative echocardiography [24]. Right heart dysfunction occurring within the first seven postoperative days from the index procedure was categorized into four stages: 0—no, 1—mild, 2 –moderate, and 3—severe right heart dysfunction. Left heart dysfunction was defined by repetitive measurements for the left ventricle function using fractional shortening (FS), Simpson uni/-biplane method, and mitral annular plane systolic excursion (MAPSE) on serial postoperative echocardiography [25, 26]. In case of an insufficient postoperative acoustic window, left ventricular function was assessed and controlled in repeated echocardiograms. Corresponding to right heart dysfunction, the left heart dysfunction was also categorized into four stages: 0—no, 1—mild, 2 –moderate, and 3—severe left heart dysfunction. Postoperative hypotension depending on the deviation from the age-related standard value (z-score -2), the need for medical and mechanical circulation support in the postoperative period was categorized into four stages: 0—no, 1—mild, 2 –moderate, and 3 –severe postoperative hypotension. The use of inotropes (epinephrine, norepinephrine, and milrinone) was defined as a continuous infusion over 24 hours postoperatively. Perioperative transfusions (erythrocytes, fresh frozen plasma, thrombocytes) were given within the complete perioperative course (intraoperatively and entire postoperative intensive care unit (ICU) stay). Major postoperative complications (e.g., pleural effusion, chylothorax, or arrhythmias) were analyzed. Procedure-related factors were defined as complications regarding the index operation (pleural effusion, chylothorax, diaphragmatic paresis, thrombosis); arrhythmias requiring permanent or transient therapy (cooling and/or amiodarone treatment, transient or permanent pacemaker) were summarized. Unscheduled re-operation or re-intervention was defined as the treatment of an anatomical residuum after the index procedure required in the postoperative course prohibiting normal recompensation and recovery. Postoperative infections were defined as the deterioration of clinical state with an elevation of laboratory parameters indicative of inflammation (leukocytosis with or without (w/-) neutrophilia with an elevation of procalcitonin w/- C-reactive protein), in combination w/- blood or wound cultures tested positive for pathogens and the need for escalating the antibiotic therapy (using broad-spectrum antibiotics). Total parenteral nutrition was defined as predominantly intravenous nutrient supply (carbohydrates, proteins, and fats) more than 24 hours postoperatively.

Laboratory blood parameters reflecting cardiac function and perfusion (brain natriuretic peptide (proBNP), troponin T (TNT), central venous saturation (CVS), postoperative lactate

level elevation), parameters reflecting liver synthesis (glutamate dehydrogenase (GLDH), albumin, and international normalized ratio (INR) for prothrombin time)) as well as liver dysfunction (aspartate transaminase (AST) and alanine transaminase (ALT)) and cholestasis (gamma-glutamyltransferase (gGT), bilirubin) and renal function (creatinine) were selected to be analyzed, and their preoperative baseline and their peak or minimum postoperative values were detected.

The trend towards normalization of altered laboratory values was calculated for each parameter as a time to event analysis. The event was defined as the time of delay from index operation to laboratory value normalization without the need for further therapy or substitution (e.g., the substitution of albumin or clotting factors).

Postoperatively liver sonography was performed in patients who maintained postoperative cholestasis. Histopathology analysis (liver biopsy) was performed in patients suffering from chronic postoperative cholestatic liver disease and being considered for liver transplantation.

## 2.3 Data analysis

Demographic, as well as pre-, intra-, and postoperative data, were obtained and analyzed for each patient. Categorical variables were presented with n (%) and analyzed using Chi-squared-test. Continuous variables were presented with mean and standard deviation and analyzed using a t-test for independent samples.

Potential risk factors indicative of the development of postoperative cardiac hepatopathy were determined by logistic regression. Bivariate and multivariate analyses were carried out by calculating the specific odds ratios and their respective 95% confidence intervals. In order to discriminate perioperative transfusion quantities (total transfusion volume of erythrocytes, fresh frozen plasma, and thrombocytes for each case), a logarithmization of the corresponding total volume for each parameter was performed. Below detection limits meaning no transfusion in this case, 0.5 was assumed to ensure error-free logarithms. Subsequently, a bivariate logistic regression was calculated based on the logarithmic transfusion values.

For comparison of time to normalization of laboratory values between non—hepatopathy group and hepatopathy group we performed Kaplan-Meier analyses and tested for significance by applying the Log rank-test.

Results were considered significant when $p < 0.05$.

Statistical analyses were performed using SPSS, Version 26 (IBM Corp., Armonk, NY, USA) and R version 4.0.2 (R Core Team. R: A Language and Environment for Statistical Computing. R Foundation for Statistical Computing. Vienna, Austria, 2020).

## 3. Results

Demographic and perioperative data are displayed in Table 1. When comparing the hepatopathy (HP) and non-hepatopathy (n-HP) group, no significant differences in preoperative risk factors (syndromic disorder, congenital heart disease, STS-EACTS category) could be detected.

In detail, the following syndromic / genetic disorders were detected: In the HP group Trisomy 21, DiGeorge syndrome, CHARGE ass., VACTERL ass., and mitochondropathy (one each). In the n-HP group: Trisomy 21 (n = 2), DiGeorge syndrome, and VACTERL ass (one ech). There was no Alagille syndrome detected.

Age at time of cardiac operation showing a wide range by analyzing patients from neonatal to toddler age differed significantly (HP: 245 +/- 289 days of life vs. n-HP: 118 +/- 118 days; p = 0.033).

According to bivariate logistic regression (Table 2), sex, cardiopulmonary bypass time, transfusion of erythrocytes, right heart dysfunction, left heart dysfunction, postoperative

**Table 1. Demographic and perioperative data.**

| | | hepatopathy group (HP) | non-heptopathy group (n-HP) | p-value |
|---|---|---|---|---|
| Patients | n | 20 | 30 | |
| Sex | male | 7 (35%) | 23 (76.7%) | |
| Age at operation | (months, mean +/- sd) (25–75% IQR) | 8 +/- 9.5 (51–318) | 3.8 +/- 3.9 (15–186) | **0.033**[*2] |
| Syndromic disorder | in total | 5 (25%) | 4 (13%) | 0.293[*1] |
| Congenital heart defect | Septal defects (VSD, AVSD w/ unbalance) | 4 | 3 | |
| | Arterial abnormalities (aortic arch hypopl. w/ VSD) | 1 | 0 | |
| | Venous abnormal. (TAPVC) | 0 | 1 | |
| | Right ventricle outflow abnormal. (ToF, w/ MAPCAs, PA/VSD, TOF abs. PV syndrome, TAC) | 7 | 7 | |
| | Left ventricle outflow abnormalities (aortic stenosis) | 1 | 0 | |
| | Abnormalities of the origin of the great arteries (simple / complex dTGA, w/ PHT, DORV (ToF-type, Taussig-Bing) | 5 | 17 | |
| | Complex cardiac abnormal. (DILV, heterotaxy syndrome) | 1 | 2 | |
| | Others (DCM) | 1 | 0 | |
| STS-EACTS Category | 1 | 0 | 0 | |
| | 2 | 5 | 4 | |
| | 3 | 3 | 18 | |
| | 4 | 11 | 8 | |
| | 5 | 1 | 0 | |
| | mean | 3.40 | 3.13 | 0.274[*3] |
| Cardiac operative proc. | CPB (min; mean +/- sd) (25–75% IQR) | 291 +/- 114 (210–373) | 202 +/- 60 (162–225) | **<0.001**[*2] |
| | AXCT (min; mean +/- sd) (25–75% IQR) | 137 +/- 57 (98–185) | 114 +/- 38 (96–127) | 0.090[*2] |
| VA-ECMO | | 7 (35%) | 4 (13%) | 0.070[*1] |
| Peritoneal dialysis | | 4 (20%) | 3 (10%) | 0.318[*1] |
| Sec. chest closure | | 15 (75%) | 15 (50%) | 0.077[*1] |
| Transfusion, periop. | erythrocytes (ml, mean) | 774 | 485 | **0.019**[*2] |
| | fresh frozen plasma (ml, mean) | 649 | 537 | 0.278[*2] |
| | thrombocytes (ml, mean) | 165 | 105 | 0.247[*2] |
| Right heart dysfunction | none (0) | 0 | 6 | |
| | mild (1) | 5 | 17 | |
| | moderate (2) | 7 | 6 | |
| | severe (3) | 8 | 1 | |
| | mean | 2.15 | 1.07 | **<0.001**[*2] |
| Left heart dysfunction | none (0) | 0 | 6 | |
| | mild (1) | 7 | 20 | |
| | moderate (2) | 10 | 0 | |
| | severe (3) | 3 | 4 | |
| | mean | 1.8 | 1.07 | **0.003** |
| Hypotension postop. | none (0) | 0 | 7 | |
| | mild (1) | 6 | 19 | |
| | moderate (2) | 11 | 4 | |
| | severe (3) | 3 | 0 | |
| | mean | 1.85 | 0.9 | **<0.001** |
| Perfusion parameters | SvO2 (min. postop, %, mean +/- sd) | 45.2 +/- 9.65 | 55.8 +/- 12.4 | **0.002**[*2] |
| | lactate (max. postop, mmol/, mean +/- sd) | 7.0 +/- 4 | 4.5 +/- 3.1 | **0.015**[*2] |
| Inotropes | use of epinephrine | 19 (95%) | 25 (83.3%) | 0.214[*1] |

*(Continued)*

**Table 1.** (Continued)

| | | hepatopathy group (HP) | non-heptopathy group (n-HP) | p-value |
|---|---|---|---|---|
| | use of norepinephrine | 6 (30%) | 5 (16.6%) | 0.265[*1] |
| | use of milrinone | 19 (95%) | 29 (96.7) | 0.768[*1] |
| Proc. related factors | pleural effusion | 14 (70%) | 7 (23.3%) | **0.001**[*1] |
| | chylothorax | 5 (25%) | 4 (13.3%) | 0.293[*1] |
| | diaphragmatic paresis | 4 (13.3%) | 2 (6.7%) | 0.155[*1] |
| | thrombosis | 1 (5%) | 4 (13.3%) | 0.336[*1] |
| | arrhythmias: | 16 (80%) | 11 (36.6%) | **0.003**[*1] |
| | • junctional ectopic tachycardia | 2 (10%) | 4 (13.3%) | |
| | • other tachycardias requiring therapy | 10 (50%) | 6 (20%) | |
| | • high grade AV block | 4 (20%) | 0 (0%) | |
| | • transient bradycardias | 0 (0%) | 1 (3.3%) | |
| | proc. related fact. (in total) | 20 (100%) | 18 (60%) | **0.001**[*1] |
| Unscheduled re-op. / re-intervent. | | 11 (55%) | 4 (13%) | **0.002**[*1] |
| Infection postop | | 15 (75%) | 8 (26.6%) | **0.001**[*1] |
| Total parent. nutrition | | 16 (80%) | 6 (20%) | **<0.001**[*1] |
| Time on ventilator | (hours, mean) | 746 | 205 | **0.001**[*2] |
| Hospitalization time | (days, mean) | 104 | 30 | **<0.001**[*2] |
| Liver transplantation LTXfollowing cardiac procedures | LTX executed | 6 (30%) | 0 | |
| | evaluation to LTX | 1 (5%) | 0 | |
| | listing to LTX | 2 (10%) | 0 | |
| Death in follow up | | 9 (45%) | 1 (3.3%) | **<0.001**[*1] |

Abbr.: abnormal.–abnormalities, abs.–absent, AVSD–atrioventricular septal defect, AXCT–aortic cross-clamp time, CPB–cardiopulmonary bypass time, CVS–central venous oxygenation, DCM–dilative cardiomyopathy, DILV–double inlet left ventricle, DORV–double outlet right ventricle, dTGA–dextro transposition of the great arteries, hypopl.–hypoplasia, JET–junctional ectopic tachycardia, LTX–liver transplantation, MAPCA–main aortopulmonary collateral artery, re-intervent.–re-intervention, re-op–re-operation, parent.–parenteral, PA/VSD–pulmonary atresia with ventricular septal defect, PD–peritoneal dialysis, periop.–perioperative, PHT–pulmonary hypertension, postop–postoperative, proc.–procedure, PV–pulmonary valve, re-op.–re-operation, re-intervent.–re-intervention, sd–standard deviation, sec.–secondary, SvO2 –central venous oxygenation, TAC–truncus arteriosus communis, TAPVC–total anomalous pulmonary venous connection, ToF–tetralogy of Fallot, VA-ECMO–venoarterial extracorporeal membrane oxygenation, VSD–ventricle septal defect, w/–with or without p-values were calculated using chi-squared-test ([*1]) for categorical variables and using t-test for independent samples ([*2]).

hypotension, minimal postoperative central venous oxygenation, postoperative lactate elevation, postoperative complications (e.g., pleural effusion, heart rhythm disorders; occurrence postop in total), unscheduled re-operation / re-intervention, infection, total parenteral nutrition, time period on the ventilator, and hospitalization time were found to be significant risk factors for developing hepatopathy. In contrast, aortic cross-clamp time, postoperative venoarterial extracorporeal membrane oxygenation (VA-ECMO), postoperative peritoneal dialysis and postoperative inotropic use (epinephrine, norepinephrine, or milrinone) did not elevate the risk for postoperative hepatopathy significantly.

Using multivariate logistic regression (Table 3) postoperative elevated lactate levels (OR 2.25; 0.006), right heart dysfunction (OR 8.98; p = 0.027), unscheduled re-operation or re-intervention (OR 86; p = 0.02) adjusted to STS-EACTS score (OR 1.61; p = 0.616) as well as sex (OR 0.37; p = 0.435) and age at operation (OR 2.27; p = 0.006) were independently, significantly associated with the development of postoperative hepatopathy.

Pre- and postoperative laboratory blood parameters indicating circulatory, renal, and hepatic function are displayed in Table 4. There were no significant differences in the

**Table 2. Bivariate risk factor analysis.**

|  | Odds ratio | 95% confidence interval | p-value |
|---|---|---|---|
| Sex: male | 0.16 | 0.04–0.55 | **0.005** |
| Age at operation | 3.42 | 1.12–15.5 | 0.060 |
| STS-EACTS category | 1.6 | 0.75–3.62 | 0.232 |
| Cardiopulmonary bypass time (CPB) | 2.07 | 1.32–3.69 | **0.005** |
| Aortic cross clamp time (AXCT) | 1.97 | 0.92–4.67 | 0.096 |
| VA-ECMO | 3.5 | 0.89–15.49 | 0.079 |
| Peritoneal dialysis | 2.25 | 0.44–12.68 | 0.326 |
| Secondary chest closure | 3.0 | 0.91–11.19 | 0.082 |
| Transfusion erythrocytes[lg] | 3.6 | 1.41–10.98 | **0.013** |
| Transfusion fresh frozen plasma[lg] | 1.51 | 0.65–3.79 | 0.350 |
| Transfusion thrombocytes[lg] | 1.05 | 0.78–1.47 | 0.750 |
| Right heart dysfunction | 5.4 | 2.32–15.88 | **<0.001** |
| Left heart dysfunction | 3.01 | 1.44–7.35 | **0.007** |
| Postoperative hypotension | 10.89 | 3.41–47.78 | **<0.001** |
| Central venous oxygenation (minimum postop) | 0.92 | 0.87–0.92 | **0.006** |
| Lactate (maximum postop) | 1.23 | 1.4–1.5 | **0.025** |
| Use of epinephrine | 3.8 | 0.55–76.04 | 0.240 |
| Use of norepinephrine | 2.14 | 0.55–8.7 | 0.270 |
| Use of milrinone | 0.66 | 0.02–17.24 | 0.770 |
| Procedure related factors (in total) | 6.91 | 1.84–25.96 | **0.004** |
| Procedure related factors: pleural effusion | 7.67 | 2.25–29.63 | **0.002** |
| Procedure related factors: cardiac arrhythmias (in total) | 6.91 | 1.97–29.15 | **0.004** |
| Unscheduled reoperation / reintervention | 7.94 | 2.15–35.01 | **0.003** |
| Infection (postop) | 8.25 | 2.39–32.92 | **0.001** |
| Total parenteral nutrition | 16 | 4.24–74.64 | **<0.001** |
| Time on ventilator | 1.08 | 1.03–1.15 | **0.009** |
| Hospitalization time | 1.05 | 1.02–1.08 | **<0.001** |

Abbr.: VA-ECMO—venoarterial extracorporeal membrane oxygenation, PD—peritoneal dialysis, postop–postoperative, lg–logarithmized

OR, 95%-CI and p-values were calculated using bivariate logistic regression

preoperative baseline values to any of the laboratory parameters analyzed (albumin differs within the normal range). Hepatopathy patients revealed significantly altered parameters for cardiac function (proBNP), liver synthesis (GLDH, INR, and albumin) and cholestasis

**Table 3. Multivariate risk factor analysis.**

|  | Odds ratio | 95% confidence interval | p-value |
|---|---|---|---|
| Sex: male | 0.37 | 0.02–4.9 | 0.435 |
| Age at operation | 2.27 | 1.4–4.9 | **0.006** |
| STS-EACTS category | 1.61 | 0.2–12.7 | 0.616 |
| Right heart dysfunction | 8.98 | 1.8–99.3 | **0.027** |
| Lactate (maximum postop) | 2.25 | 1.4–4.7 | **0.006** |
| Unscheduled reoperation / reintervention | 86 | 3.4–8691.8 | **0.02** |

Abbr.: postop–postoperative

OR, 95%-CI and p-values were calculated using multivariate logistic regression

**Table 4. Parameters of circulatory and hepatologic function.**

| lab value | | hepatopathy group (HP) | non-hepatopathy group (n-HP) | p-value |
|---|---|---|---|---|
| | | mean (+/- sd) | mean (+/- sd) | |
| **ProBNP** (ng/l) | preop | 8996 (+/- 10210) | 7884 (+/- 10432) | 0.711 |
| (norm: < 155 ng/l) | postop | 123267 (+/- 298174) | 32556 (+/- 29232) | **0.044** |
| **TNT** (pg/ml) | preop | 39 (+/- 33) | 50.69 (+/- 52) | 0.4 |
| (norm: < 14 pg/ml) | postop | 6507 (+/- 3436) | 5758 (+/- 3315) | 0.435 |
| **Creatinine** (mg/dl) | preop | 0.4 (+/- 0.1) | 0.4 (+/- 0.1) | 0.816 |
| (norm: < 1.2 mg/dl) | postop | 1.3 (0.5) | 0.7 (+/- 0.3) | **<0.001** |
| **GLDH** (U/l) | preop | 8 (+/- 5) | 11 (+/- 9) | 0.325 |
| (norm: < 5 U/l) | postop | 231 (+/- 185) | 99 (+/- 214) | **0.045** |
| **INR** (%) | preop | 1.1 (+/- 0.1) | 1.1 (+/- 0.2) | 0.457 |
| (norm: 0.85–1.15) | postop | 2.8 (1.4) | 1.6 (0.5) | **<0.001** |
| **Albumin** (g/l) | preop | 37 (+/- 4) | 32 (+/- 6) | **0.01** |
| (norm: 34–50 g/l) | postop | 23 (+/- 4) | 31 (+/- 6) | **0.003** |
| **Bilirubin** (mg/dl) | preop | 1.3 (+/- 2.0) | 1.65 (+/- 2.4) | 0.557 |
| (norm: 0.2–1 mg/dl) | postop | 7.8 (+/- 6.8) | 2.4 (+/- 2.5) | **0.005** |
| **AST** (U/l) | preop | 41 (+/- 16) | 47 (+/- 28) | 0.390 |
| (norm: 15–41 U/l) | postop | 602 (+/- 601) | 355 (+/- 670) | 0.206 |
| **ALT** (U/l) | preop | 26 (+/- 17) | 29 (+/- 15) | 0.553 |
| (norm: 8–45 U/l) | postop | 426 (+/- 555) | 153 (+/- 361) | 0.072 |
| **gGT** (U/l) | preop | 97 (+/- 162) | 34 (+/- 28) | 0.058 |
| (norm: 6–30 U/l) | postop | 2120 (+/- 1472) | 22 (+/- 169) | **0.002** |

Abbr.: proBNP–pro-brain natriuretic peptide, TNT–troponin T, INR–international normalized ratio for prothrombin time, GLDH–glutamate dehydrogenase, AST–aspartate transaminase, ALT–alanine transaminase, gGT–gamma-glutamyltransferase, preop–preoperative, postop–postoperative, sd–standard deviation

p-values were calculated using t-test

(bilirubin, gGT). Creatinine as a renal function parameter was significantly higher in the hepatopathy group postoperatively (but minimally above the normal range).

Kaplan-Meier analysis (Fig 1) showed a significantly reduced probability for normalization of proBNP, TNT, GLDH, INR, albumin, bilirubin, AST, ALT, and gGT in the HP group. After an interval of approx. 1.5–2 months from the index operation, none of the values normalized and did not reveal any tendency towards normalization in the longer follow-up. For creatinine, however, a complete normalization is seen with an interval of approx. 1.5 months to the index operation.

All patients of the HP group were subjected to liver sonography, which reveals pathologic parenchymal structure and perfusion findings. Liver biopsies were conducted in 10/20 (50%) hepatopathy patients. In all 10/10 (100%) liver biopsies, histopathology analysis showed severe liver cirrhosis (Fig 2). Centrilobular necrosis was seen in 2/10 (%) liver biopsies. Moreover, liver histopathology analysis showed secondary sclerosing cholangitis (SSC) in 4/10 (40%).

## 4. Discussion

The impact of hepatopathy after complex congenital heart surgery on morbidity and mortality was cumbersome. In the postoperative course, 45% of CHD patients with hepatopathy and irreversible multiorgan dysfunction died. 30% of patients already had undergone liver transplantation. Finally, 15% of patients were evaluated or listed for liver transplantation in the next future.

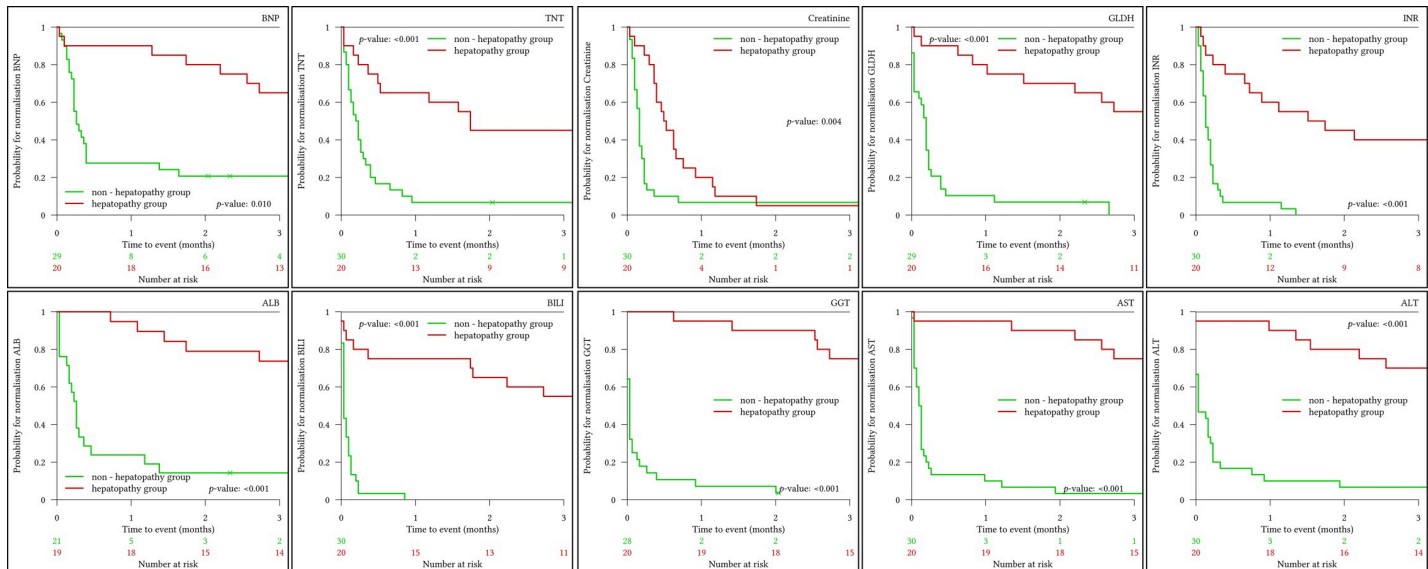

**Fig 1. Laboratory trend normalization analysis.** Kaplan Meier analysis to calculate the probability of value normalization for the following laboratory parameters: BNP–pro-brain natriuretic peptide, TNT–troponin T, creatinine, GLDH–glutamate dehydrogenase, INR–international normalized ratio for prothrombin time, ALB–albumin, BILI–bilirubin, GGT–gamma-glutamyltransferase, AST–aspartate transaminase and ALT–alanine transaminase. Time to event describes the time period for complete laboratory value normalization.

Following complex congenital cardiac surgery, organ dysfunction is widely recognized regarding renal or neurologic function. Nevertheless, little is known about progressive hepatopathy in this setting. Hyperbilirubinemia and elevation of serum transaminases are prognostic factors for hepatic complications [27, 28]. A recent study correlated hyperbilirubinemia as a sign of liver dysfunction following neonatal heart surgery and the need for ECMO and total parental nutrition as the main risk factors in the postoperative course [29].

In our series, factors reflecting hemodynamic depression (lactate elevation, CVS decrease) and right heart dysfunction were found as independent risk factors for cardiac hepatopathy.

After complex cardiac surgery, postoperative low cardiac output syndrome (LCOS) with reduced left heart function, postoperative hypotension, low CVS and lactate elevation is common and likely an important factor in the formation of cardiac hepatopathy. It is well known that SvO2 is an objective parameter of oxygen delivery and outcome in single and biventricular

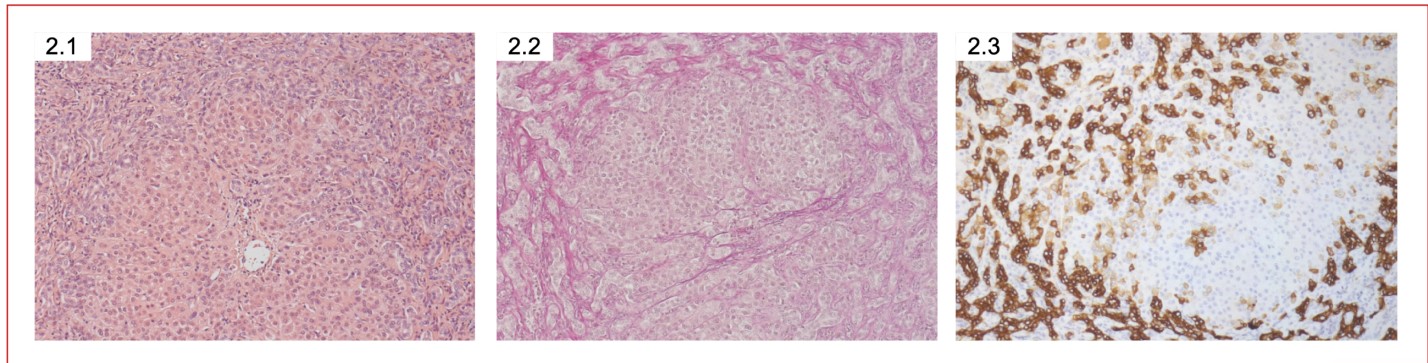

**Fig 2. Histopathology of the liver in cardiac hepatopathy.** Severe liver cirrhosis and cholestasis with an increase of bile ductus were shown using hematoxylin and eosin stain (2.1, 2.2) and immunohistochemical staining for cytokeratin 7 (2.3).

circulation and septic shock [30, 31]. In addition, the postoperative peak lactate level as another indicator for LCOS was shown to be a risk factor for hepatopathy in our analysis. This observation is interesting in so far as the liver contributes about 60% to lactate metabolism [32].

Right heart dysfunction was previously described as an important entity for the development of cardiac hepatopathy [11]. In line with that, our data support this finding. The subtype of CHD might also influence the formation of cardiac hepatopathy. In our study, there is a tendency towards hepatopathy in patients suffering from right heart pathologies. The mechanisms in right heart failure and corresponding hepatopathy, or especially in the situation of failing Fontan circulation, nowadays are better understood [12, 13, 33, 34]. Elevated central venous pressure in the inferior vena cava leads to liver sinusoid dilatation, stretching, and stimulation of satellite cells also described as "congestive hepatopathy," ending in fibrotic reaction [14, 17]. Low cardiac output diminishes portal venous circulation, limits hepatic arterial buffer response, and reduces hepatic perfusion. These factors contribute to liver hypoxia and ischemic injury, which also may lead to liver fibrosis [34].

Other authors separate the term "congestive hepatopathy" caused by chronic accumulation of liver damage associated with hepatic venous congestion from the term "ischemic or hypoxic hepatitis" caused by left-sided forward failure [13, 35]. Histopathologic findings from our patients indeed support the differentiation in hypoxic liver injury with the detection of necrosis of centrilobular hepatocytes and congestive hepatopathy with centrilobular congestion and sinusoidal dilatation, atrophy of centrilobular hepatocytes, and a variable degree of fibrosis [13]. Histopathologic analyses of the biopsies of our cohort who underwent liver transplantation revealed centrilobular necrosis as well as sinusoidal atrophy with a severe degree of fibrosis, which underlines both possible mechanisms of liver injury. Our thesis of right and left heart failure as the main cause for cardiogenic hepatopathy is also underlined by published data of an animal model, in which low cardiac output induced by right heart failure after acute elevation of right ventricle outflow pressure caused by tight pulmonary artery banding leads to hepatic fibrosis [36].

Secondary sclerosing cholangitis (SSC) is another histopathologic finding in specimens with chronic biliary obstruction. In at least 3/10 patients, SSC was detected. In recent literature, general hemodynamic depression in context with cardiothoracic surgery is discussed as a leading cause of SSC [19]. Our data support these findings. Secondary sclerosing cholangitis in critically ill patients is associated with poor prognosis and often rapidly progressing to liver cirrhosis with the urgent need for liver transplantation [15]. In our cohort, we found a prolonged elevation of markers for intrahepatic cholestasis (gGT and bilirubin), transaminases elevation, and a prolonged decrease of synthesis parameters (albumin, prothrombin time) correlating significantly with a high risk for postoperative cardiac hepatopathy (Fig 1).

Factors reflecting the complexity of the surgical procedure, like the time on CPB, the volume of perioperative erythrocytes transfusion, and procedure associated complications (e.g., pleural effusion, chylothorax or heart rhythm disorders) with the need for further interventions (e.g., pacemaker implantation, pleurodesis) influenced the risk for cardiac hepatopathy in our cohort. On multivariate analysis, relevant postoperative anatomical residuals requiring re-operation or re-intervention (e.g., re-operation on vessel stenosis or interventional stenting) significantly increased the risk for cardiac hepatopathy.

In line with our results, risk factors reflecting the extent of postoperative morbidities like the duration of mechanical ventilation, the need for parenteral nutrition, hospitalization time, and ICU stay have been described by other groups [37]. Moreover, the development of postoperative infections can also be involved in the formation of cardiac hepatopathy. Our data show a significantly higher ratio of infections in patients with hepatopathy. Due to its retrospective

character, we could not clarify whether infections caused cardiac hepatopathy or whether patients suffering from postoperative hepatic dysfunction, in general, are at increased risk for postoperative infections.

The observation that the age at the time of surgery influences the incidence of hepatopathy in the multivariate analysis could be related to the heterogenic age distribution of this cohort. In terms of heterogeneity, there were a number of syndromic patients in both groups, but without statistical difference between both groups. Furthermore, there were no patients with Alagille syndrome, known to have a liver impact [38–40].

This study is weakened by its retrospective character, the limited number of cases, and a single-center experience. Furthermore, both the age and the underlying congenital heart defects of the hepatopathy patients shown here are heterogeneous, which makes a direct comparison with the control group challenging. Therefore, a statistically unassailable matched-control design could not be applied. Nevertheless, we believe that we could identify significant risk factors for the development of post-cardiac surgery hepatopathy in pediatric patients with congenital heart defects. Further multi-institutional studies are required to confirm and interpret those findings and may provide means of prevention.

## 5. Conclusion

Postoperative cardiac hepatopathy is probably a widely overlooked problem after surgery for congenital heart disease. The presented data indicates a multifactorial genesis related to severe postoperative right heart dysfunction with hepatic and biliary congestion yielding in a so-called "congestive hepatopathy" and low cardiac output states leading to "ischemic hepatopathy". All perioperative factors leading to such a condition like anatomic residuals, prolonged bypass times, and extended intensive care unit treatment contribute to this relevant problem. An important message of this study is the fact that the lacking normalization of parameters of liver synthesis and cholestasis is a significant indicator of an upcoming liver sequel and should be monitored closely.

## Supporting information

**S1 Material.**
(ZIP)

## Author Contributions

**Conceptualization:** Torben Kehl, Daniel Biermann, Andrea Briem-Richter, Gerhard Schoen, Urda Gottschalk.

**Data curation:** Torben Kehl, Gerhard Schoen.

**Formal analysis:** Torben Kehl, Gerhard Schoen.

**Investigation:** Torben Kehl, Daniel Biermann, Andrea Briem-Richter, Jakob Olfe, Urda Gottschalk.

**Software:** Torben Kehl.

**Supervision:** Joerg S. Sachweh, Rainer Kozlik-Feldmann.

**Validation:** Daniel Biermann, Gerhard Schoen, Joerg S. Sachweh, Urda Gottschalk.

**Visualization:** Torben Kehl, Hansjoerg Schaefer.

**Writing – original draft:** Torben Kehl, Daniel Biermann, Andrea Briem-Richter, Urda Gottschalk.

**Writing – review & editing:** Daniel Biermann, Andrea Briem-Richter, Gerhard Schoen, Jakob Olfe, Joerg S. Sachweh, Lutz Fischer, Hansjoerg Schaefer, Rainer Kozlik-Feldmann, Urda Gottschalk.

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
