## [Decision Letter · Decision Letter 0]

16 Nov 2020

PONE-D-20-31616

Impact of Hepatopathy in Pediatric Patients after Surgery for Complex Congenital Heart Disease

PLOS ONE

Dear Dr. Kehl,

Thank you for submitting your manuscript to PLOS ONE. After careful consideration, we feel that it has merit but does not fully meet PLOS ONE’s publication criteria as it currently stands. Therefore, we invite you to submit a revised version of the manuscript that addresses the points raised during the review process.

The article has been reviewed by 2 expert reviewers who found some interesting points and several weaknesses. I think the paper requires significant revisions before next submission and the authors will have to thoroughly address all reviewers requests.

We look forward to receiving your revised manuscript.

Kind regards,

Zaccaria Ricci

Academic Editor

PLOS ONE

Journal Requirements:

2. Thank you for stating in the text of your manuscript " The local ethics committee waived approval of the study due to the retrospective and anonymous analysis performed (WF-055/20; ethics

committee of the medical association Hamburg, Germany)." Please clarify whether the committee waived the need for ethics approval, informed consent, or both. Please also clarify whether all patient data was de-identified before researchers had access to it. Please add all of this information to your manuscript text and ethics statement in the online submission form.

3. We note that table 1 give a p-value of .274 for STS-EACTS Category and is bolded. However, we note that only p-values less than 0.05 are bolded inthe rest of the table. Please ensure that you have provided the correct p-value or that you have bolded correctly.

4. In your Methods section, please provide additional information about the participant recruitment method and the demographic details of your participants. Please ensure you have provided sufficient details to replicate the analyses such as:

a) a description of any inclusion/exclusion criteria that were applied to participant recruitment

b) a statement as to whether your sample can be considered representative of a larger population

"NO"

6. We note that you have indicated that data from this study are available upon request. PLOS only allows data to be available upon request if there are legal or ethical restrictions on sharing data publicly. For information on unacceptable data access restrictions, please see http://journals.plos.org/plosone/s/data-availability#loc-unacceptable-data-access-restrictions.

Reviewers' comments:

Reviewer's Responses to Questions

**Comments to the Author**

1. Is the manuscript technically sound, and do the data support the conclusions?

Reviewer #1: Yes

Reviewer #2: Yes

2. Has the statistical analysis been performed appropriately and rigorously? 

Reviewer #1: No

Reviewer #2: Yes

3. Have the authors made all data underlying the findings in their manuscript fully available?

Reviewer #1: No

Reviewer #2: Yes

4. Is the manuscript presented in an intelligible fashion and written in standard English?

Reviewer #1: Yes

Reviewer #2: Yes

5. Review Comments to the Author

Reviewer #1: Kehl and coworkers conducted a study on hepatopathy secondary to pediatric cardiac surgery describing its predictors and its outcomes. The study is original and the results interesting even if the cohort is small in a relatively long period of time (20 patients in 8-9 years)

I have a series of concerns to be addressed in this paper

-the authors state they compared their patients "to a control group consisting of 30 patients with a comparable initial cardiac anatomy and pathophysiology" but I am not sure this is consistent with the methodology of a matched-controlled study or, better, a propensity study. The authors should significantly revise their statistical methodology otherwise, as is, the main message emerging from this study could be: "complicated surgery is a risk factor for hepathopathy" which is quite a tautologic message

-the other MAJOR aspect the authors should fix is the definition of "hepathopathy": this should be explicitly proposed and detailed or, better, some standard definition of hepathopathy should be described here.

-I have to caution the authors that some important information included in the abstract like mortality, liver transplantation rate and "relevant postoperative anatomical residuum" (by the way what is meant by "relevant") are not reported in the main text

-the authors should specify how long the follow up was prolonged and what is the postoperative window all the clinical and lab data are referred to

-why is not "left ventricular function" (which is ALSO cited in the abstract) analyzed and presented?

-why is not blood pressure analyzed and presented?

-to this end, please specify how the kaplan meyer curves were created: what event is referred the "time to event"?

-I think the authors should add information about renal function since the item "peritoneal dialysis" is very poor to describe it and it frequently occurs in liver dysfunction

-the authors should specify how table3 (multivariate analysis) was built

-why and when the hepatic biopsies where conducted? what about the remaining patients?

-what the term infection in the table is referred to? blood? any infection? why do the authors refer to this variable as "severe infection" in the text?

-the limitations section is very poor and the author should honestly describe the multiple weaknesses of this exploratory study (other than the small sample size)

Reviewer #2: Dear Editor,

It is was a pleasure to review this study on a very interesting and probably underestimated subject. The lack of clinical experience on the hepatic involvement and its outcome after cardiac surgery might be of clinical and scientific interest to readers of this journal despite the retrospective and single center nature of the study. The paper is well organized, fully detailed, with intelligible fashion and written in standard English.

However, I have the following specific remarks and suggestions:

1) Authors present 20 patients who developed hepatic dysfunction vs 30 patients (controls) who did not. These 50 patients were consecutive? What was the overall series of cardiac surgery during the study time (2011-2019)? Even if it is a single center experience, what was the overall prevalence of hepatic dysfunction?

2) Age at cardiac surgery shows wide range (p 0.033) between the two populations, why they are not age-matched? In the hepatopathy group patients are older, is it possible to consider the time (even if short) from CHD diagnosis to surgery a risk for hepatic dysfunction? Please comment on this

3) It would be fair to presents the liver parameter (laboratory and ultrasound) for both case/controls before surgery to rule out or address a previous hepatic involvement

4) Syndromic disorders account for 25% in HP group. Were there any patients with Alagille syndrome and related known chronic liver disease?

5) Authors underline how right heart dysfunction is a well-recognized entity for the development of cardiac hepatopathy and their data support this finding. I would like to ask for clarification about Table 1 (page 8):

Right heart dysfunction seems to occur with a different degree in the same proportion between the two groups (19/20 vs 30/30). Please a comment.

6) Table 4 reports parameters of hepatologic function:

- What was the peak bilirubin level for both groups?

- Do albumin and PT were supported?

- Adding INR could be useful to calculate PELD score. PELD score (and also FV%) should be included given the severity of the liver disease as reported by authors. It is remarkable the rate of 40% of patients evaluated for liver transplantation and then performed in 30%. These rates are surprisingly high considering the clinical setting and age of patients, such that I believe this aspect needs more details in both results and discussion adding data and other experience from literature.

Page 3 (line 69) : Liver dysfunction in Fontan failure is uniformly known. Noteworthy, the need for liver transplant in Fontan failure seems low (mainly as a treatment of HCC or when combined heart-liver Tx come to discussion) than what was reported by authors in their case series and after a longer follow-up. Time of FU as a variable associated with the degree of liver disease. Comment please.

Liver biopsy was performed in 50% of HP groups, what was the indication for risky liver histology in this group? Authors recognized how acute hepatic dysfunction is mainly secondary to cardiac condition (right and left hit), as a conclusion of their study would they still suggest liver biopsy in this group?

Did the authors employ extracorporeal filtration (plasmapheresis or CytoSorb) for any of the HE patients ?

7) Line 316 : “strength of our study is the very detailed information”, it seems too emphatic probably to be reformulated.

6. PLOS authors have the option to publish the peer review history of their article (what does this mean?). If published, this will include your full peer review and any attached files.

Reviewer #1: No

Reviewer #2: No

---

## [Author Response · Author response to Decision Letter 0]

3 Feb 2021

Response to Reviewers

We would like to thank the reviewers for the extensive and thorough evaluation of our work. In the following, we would like to take a firm stand on the individual points mentioned. All review-ers’ points have been considered in the revised manuscript.

Reviewer #1:

1. „the authors state they compared their patients "to a control group consisting of 30 pa-tients with a comparable initial cardiac anatomy and pathophysiology" but I am not sure this is consistent with the methodology of a matched-controlled study or, better, a propensity study. The authors should significantly revise their statistical methodology otherwise, as is, the main message emerging from this study could be: "complicated surgery is a risk factor for hepatopathy" which is quite a tautologic message”

Answer from the authors:

The method of obtaining the control group was narrowly limited due to the small num-ber of cases. Both matched pairs selection and drawing by propensity score require that a sufficiently large pool of data is available. As this was not the case, we decided to establish equality between the two groups by initial cardiac anatomy and pathophys-iology. The extent to which structural equality was also achieved with respect to other characteristics can be seen in Table 1. It is a limitation of the study that it is not a pro-spective study, but the two groups were selected according to the outcome "hepatopathy".

2. “the other MAJOR aspect the authors should fix is the definition of "hepatopathy": this should be explicitly proposed and detailed or, better, some standard definition of hepatopathy should be described here.”

Answer from the authors:

This comment has now been taken into account as follows: “Cardiac hepatopathy was defined based on published definitions as persistent postoperative cholestasis with an elevation of specific laboratory cholestasis parameters (bilirubin, gamma-glutamyltransferase (gGT)), impaired liver synthesis (reduced production of albumin and/or coagulation factors) with or without liver sonographic and histopathological changes in the postoperative course”. Histopathologies were performed only in pa-tients who developed chronic cholestasis (see line 159 ff.). Some of the patients showed a fatal course and could not be submitted to a dedicated liver diagnostic.

3. “I have to caution the authors that some important information included in the abstract like mortality, liver transplantation rate and "relevant postoperative anatomical residu-um" (by the way what is meant by "relevant") are not reported in the main text”

Answer from the authors:

All variables presented in the abstract are now also mentioned in the main text and more precisely defined in the methods (line 89 ff.) and results sections (see line 184 ff.), and table 1-3.

A postoperative anatomical residuum was defined as a state which needs a re-operation or re-intervention named as “unscheduled re-operation or re-intervention” in the following postoperative course.” (see line 137 ff.). We removed the term “relevant” form the abstract.

4. “the authors should specify how long the follow up was prolonged and what is the postoperative window all the clinical and lab data are referred to”

Answer from the authors:

The follow-up period was 4 months to 4 years. This information was also added to the manuscript (see line 103). The monitoring period differed due to heterogeneous courses. Patients who developed a severe and chronic course were listed and man-aged for liver transplantation; other patients died in the course after surgical interven-tion (see table 1).

5. “why is not "left ventricular function" (which is ALSO cited in the abstract) analyzed and presented?”

Answer from the authors:

Based on this justified comment, the corresponding echocardiograms were re-examined and left heart dysfunction was categorized according to right heart dysfunc-tion (see lines 119 ff.).

6. “why is not blood pressure analyzed and presented?”

Answer from the authors:

Arterial blood pressure was analyzed and the results were included in the results sec-tion of the manuscript. To take the heterogeneous age of the patients (different blood pressure values) into account, we categorized blood pressure depression postopera-tively (please see 3.2 Definitions, line 126 ff.).

7. “to this end, please specify how the kaplan meyer curves were created: what event is referred the "time to event"?”

Answer from the authors:

The Kaplan Kaplan-Meier estimator has to be used whenever censored data is in-volved. In this case, the event is the normalization of a laboratory parameter and the time to event is the time that has elapsed until the normalization has occurred. The corresponding test for the difference between two groups is the log-rank test.

8. “I think the authors should add information about renal function since the item "perito-neal dialysis" is very poor to describe it and it frequently occurs in liver dysfunction”

Answer from the authors:

To answer this justified point of criticism we now describe postoperative renal function using creatinine as a renal function marker (see revised table 4). There was a signifi-cant increase in creatinine postoperatively in the hepatopathy patients studied. How-ever, the increase was only minimally above the normal range. In addition, we ana-lyzed the probability of normalization of creatinine with time from surgery (time-to-event analysis, see Figure 1). Creatinine was the only laboratory parameter analyzed that showed normalization less than 2 months after the index operation. We therefore do not see renal dysfunction as the leading problem in contrast to hepatopathy.

9. “the authors should specify how table 3 (multivariate analysis) was built” 

Answer from the authors:

The multivariate analysis models the probability of hepatopathy occurrence. For this purpose, we used a logistic regression model. The dependent variable was "hepatopathy" and the independent variables were selected based on preliminary con-siderations. Because of the limited number of cases, we did not use statistical variable selection procedures. All analyses were adjusted for sex and age, regardless of whether age or sex had a significant effect on outcome.

10. “why and when the hepatic biopsies were conducted? what about the remaining pa-tients?”

Answer from the authors:

Liver biopsies were only performed in patients who developed chronic end-stage liver dysfunction to guide further treatment. The indication was confirmed in our interdisci-plinary pediatric liver board (pediatric hepatology, liver transplant surgery, general pe-diatrics). 

11. “what the term infection in the table is referred to? blood? any infection? why do the authors refer to this variable as "severe infection" in the text?”

Answer from the authors:

This comment has been recognized as follows: “Postoperative infections were defined as the deterioration of clinical state with an elevation of laboratory parameters indica-tive of inflammation (leukocytosis with or without (w/-) neutrophilia with an elevation of procalcitonin w/- C-reactive protein), in combination w/- blood or wound cultures tested positive for pathogens and the need for escalating the antibiotic therapy using broad-spectrum antibiotics.” (see line 140 ff.). The term „severe infection” was removed from the manuscript.

12. “the limitations section is very poor and the author should honestly describe the multi-ple weaknesses of this exploratory study (other than the small sample size)”

Answer from the authors:

The limitation section has been rewritten and now considers all limitations of our analy-sis, particularly in terms of methodology and statistics. In detail, the age and the under-lying congenital heart defects of the hepatopathy patients are heterogeneous, making a direct comparison with the control group challenging. Therefore, a statistically unas-sailable matched-control design could not be applied. Nevertheless, we believe that we could identify significant risk factors for the development of post-surgical hepatopathy in pediatric patients with congenital heart defects.

Reviewer #2:

1. “Authors present 20 patients who developed hepatic dysfunction vs 30 patients (con-trols) who did not. These 50 patients were consecutive? What was the overall series of cardiac surgery during the study time (2011-2019)? Even if it is a single-center ex-perience, what was the overall prevalence of hepatic dysfunction?”

Answer from the authors:

We added our average number of pediatric cardiac operations per year to the manu-script (average 280 pediatric cardiac operations per year in the period 2011-2019, please compare 3.1 Definitions, line 96 ff). We included all patients in whom liver dys-function occurred during this period in our analysis.

2. “Age at cardiac surgery shows wide range (p 0.033) between the two populations, why they are not age-matched? In the hepatopathy group patients are older, is it pos-sible to consider the time (even if short) from CHD diagnosis to surgery a risk for he-patic dysfunction? Please comment on this”

Answer from the authors:

Because of the small number of cases, we could select the control group only accord-ing to initial cardiac anatomy and pathophysiology and could not consider age in the drawing. For this reason, we adjusted for age in the multivariate model. That is, all re-ported odds ratios are corrected for any age difference.

3. “It would be fair to presents the liver parameter (laboratory and ultrasound) for both case/controls before surgery to rule out or address a previous hepatic involvement”

Answer from the authors:

Table 4 was supplemented by the preoperative initial values of all analyzed laboratory values. We saw no significant differences in the laboratory values studied preopera-tively comparing the two groups. Liver sonographies were performed preoperatively in all newborns as part of the organ clarification. None of the patients mentioned here showed preoperative sonographic abnormalities. Since no abnormal liver values were found preoperatively in the entire cohort, liver sonography was not performed in older patients because there was no initial suspicion.

Postoperatively, liver sonography was only performed in patients with signs of choles-tasis (please see line 159 ff.).

4. “Syndromic disorders account for 25% in HP group. Were there any patients with Ala-gille syndrome and related known chronic liver disease?”

Answer from the authors:

In the revised manuscript, we present the syndromic disorders in the results section. To note, there was no patient with Alagille syndrome (compare line 189 ff.).

5. “Authors underline how right heart dysfunction is a well-recognized entity for the de-velopment of cardiac hepatopathy and their data support this finding. I would like to ask for clarification about Table 1 (page 8):

Right heart dysfunction seems to occur with a different degree in the same proportion between the two groups (19/20 vs 30/30). Please a comment.”

Answer from the authors:

Table 1 was revised and corrected. The average degree of right heart dysfunction dif-fered significantly with hepatopathy patients showing a more depressed postoperative right heart function.

6. “Table 4 reports parameters of hepatologic function: What was the peak bilirubin level for both groups? Do albumin and PT were supported? Adding INR could be useful to calculate PELD score. PELD score (and also FV%) should be included given the se-verity of the liver disease as reported by authors. It is remarkable the rate of 40% of patients evaluated for liver transplantation and then performed in 30%. These rates are surprisingly high considering the clinical setting and age of patients, such that I be-lieve this aspect needs more details in both results and discussion adding data and other experience from literature.”

Answer from the authors:

We have redesigned Table 4: Preoperative baseline values, postoperative maximum changes, and normal range are now shown for all laboratory parameters. The mean pre- and postoperative bilirubin level is also presented in table 4. For both groups there was no significant difference preoperatively but a significant difference postoperatively was observed (see table 4). Patients who showed an albumin or coagulation factor de-ficiency in the postoperative course were substituted according to the protocol of our pediatric intensive care unit. The period for normalization of laboratory parameters was determined as follows: Period between the time of surgery and almost complete laboratory value normalization without the need for therapy or substitution. We added INR for the analysis with preoperative mean, postoperative mean peak and time period for normalization. We did not use factor V in our analysis. We just have factor V values for those patients evaluated for liver transplantation. We did not include the PELD score in our analysis because it was calculated only for patients evaluated for trans-plantation. All patients were listed for liver transplantation at a PELD score of 28.

7. “Page 3 (line 69): Liver dysfunction in Fontan failure is uniformly known. Noteworthy, the need for liver transplant in Fontan failure seems low (mainly as a treatment of HCC or when combined heart-liver Tx come to discussion) than what was reported by au-thors in their case series and after a longer follow-up. Time of FU as a variable asso-ciated with the degree of liver disease. Comment please.”

Answer from the authors:

Chronic liver dysfunction is a well-known long-term complication in patients with Fon-tan circulation. However, the need for liver transplantation is a rarity in the literature. Our patients differ from the Fontan population in the etiology of the hepatopathy sup-porting the differentiation in hypoxic liver injury and congestive hepatopathy. The au-thors believe that, to date, hepatopathy after corrective cardiac surgery is an underre-ported complication. 

8. “Liver biopsy was performed in 50% of HP groups, what was the indication for risky liver histology in this group? Authors recognized how acute hepatic dysfunction is mainly secondary to cardiac condition (right and left hit), as a conclusion of their study would they still suggest liver biopsy in this group?

Answer from the authors:

Liver biopsy was performed in patients with clinical, laboratory, or sonographic signs of advanced liver fibrosis or end-stage liver disease to characterize liver histology. The indication for liver biopsy was confirmed in our interdisciplinary hepatic board (pe-diatric hepatology, pediatric transplantation surgery, general pediatrics). Patients with impaired coagulation at the time of liver biopsy were substituted with coagulation fac-tors according to the protocol of our center. Furthermore, the authors think that liver histology is very useful to support the decision-making process regarding the indica-tion for liver transplantation.

9. Did the authors employ extracorporeal filtration (plasmapheresis or CytoSorb) for any of the HE patients?

Answer from the authors:

No, we did not use extracorporeal filtration (plasmapheresis or CytoSorb) in these pa-tients reported here. Potentially, this is an interesting approach in the care of such pa-tients. Further studies would be necessary.

13. “Line 316: “strength of our study is the very detailed information”, it seems too emphat-ic probably to be reformulated.”

Answer from the authors:

The limitation section of the manuscript has been rewritten and now takes more ac-count of the existing limitations of our analysis.

---

## [Decision Letter · Decision Letter 1]

5 Mar 2021

Impact of Hepatopathy in Pediatric Patients after Surgery for Complex Congenital Heart Disease

PONE-D-20-31616R1

Dear Dr. Kehl,

We’re pleased to inform you that your manuscript has been judged scientifically suitable for publication and will be formally accepted for publication once it meets all outstanding technical requirements.

Kind regards,

Zaccaria Ricci

Academic Editor

PLOS ONE

Additional Editor Comments (optional):

the authors conducted a good job in revising their manuscript

Reviewers' comments:

Reviewer's Responses to Questions

**Comments to the Author**

1. If the authors have adequately addressed your comments raised in a previous round of review and you feel that this manuscript is now acceptable for publication, you may indicate that here to bypass the “Comments to the Author” section, enter your conflict of interest statement in the “Confidential to Editor” section, and submit your "Accept" recommendation.

Reviewer #1: All comments have been addressed

Reviewer #2: All comments have been addressed

2. Is the manuscript technically sound, and do the data support the conclusions?

Reviewer #1: Yes

Reviewer #2: Yes

3. Has the statistical analysis been performed appropriately and rigorously? 

Reviewer #1: Yes

Reviewer #2: Yes

4. Have the authors made all data underlying the findings in their manuscript fully available?

Reviewer #1: Yes

Reviewer #2: Yes

5. Is the manuscript presented in an intelligible fashion and written in standard English?

Reviewer #1: Yes

Reviewer #2: Yes

6. Review Comments to the Author

Reviewer #1: nice revision. all the queries have been addressed.

Reviewer #2: I read with interest all the answer provided by the authors. All the critical points seem to addressed and overall the manuscript has a better shape.

Language sounds standard english to me.

Statistical analysis improoved significantly.

The major finding in this study is the severity of liver disease following cardiac surgery requiring liver transplantation. This incidence (in children) is not commonly known or shared with other centers experience thus a more detailed review of the literature providing an hypotetic explication could have been added to the discussion.

7. PLOS authors have the option to publish the peer review history of their article (what does this mean?). If published, this will include your full peer review and any attached files.

Reviewer #1: No

Reviewer #2: No

---

## [Editor Report · Acceptance letter]

17 Mar 2021

PONE-D-20-31616R1 

Impact of Hepatopathy in Pediatric Patients after Surgery for Complex Congenital Heart Disease 

Dear Dr. Kehl:

I'm pleased to inform you that your manuscript has been deemed suitable for publication in PLOS ONE. Congratulations! Your manuscript is now with our production department. 

Kind regards, 

on behalf of

Dr. Zaccaria Ricci 

Academic Editor

PLOS ONE